# Study on the Effect of Ultraviolet Absorber UV-531 on the Performance of SBS-Modified Asphalt

**DOI:** 10.3390/ma15228110

**Published:** 2022-11-16

**Authors:** Li Liu, Leixin Liu, Zhaohui Liu, Chengcheng Yang, Boyang Pan, Wenbo Li

**Affiliations:** National Engineering Laboratory for Highway Maintenance Technology, School of Traffic and Transportation Engineering, Changsha University of Science & Technology, Changsha 410114, China

**Keywords:** road engineering, UV aging, UV absorber, high and low-temperature performance, infrared spectrum

## Abstract

Asphalt pavements at high altitudes are susceptible to aging and disease under prolonged action of UV light. To improve their anti-ultraviolet aging performance, UV-531/SBS-modified asphalts with UV-531 dopings of 0.4%, 0.7%, and 1.0% were prepared by the high-speed shear method, and the effect of UV-531 on the conventional performance of SBS-modified asphalt before aging was studied by needle penetration, softening point and 5 °C ductility tests. The high- and low-temperature rheological properties of UV-531/SBS-modified asphalt before and after aging were also analyzed by high temperature dynamic shear rheology test and low-temperature glass transition temperature test. Finally, the effect of UV-531 on the anti-aging performance of SBS-modified asphalt was evaluated by three methods, including rutting factor ratio, viscosity aging index, and infrared spectroscopy. The results show that with the increase of UV-531 doping, the needle penetration and 5 °C ductility show an increasing trend, but the effect on the softening point is small. The high temperature stability of SBS-modified asphalt is not much affected by the addition of UV-531, and the low-temperature stability is improved, and when 0.7% UV absorber is added, SBS-modified asphalt shows better low-temperature performance. The results of all three evaluation methods show that the addition of UV-531 significantly improved the anti-UV aging performance of SBS-modified asphalt, with the amount of 0.7% providing the asphalt with the best anti-UV aging performance. The results of the study can provide an important reference for improving the anti-ultraviolet aging performance of SBS-modified asphalt.

## 1. Introduction

When using asphalt mixes, the process of ultraviolet radiation induced by asphalt UV aging cannot be neglected, since asphalt mixes are prone to aging under heat or natural conditions [1,2]. Since old asphalt is more susceptible to spalling, potholes, and other pavement diseases, it is imperative to find solutions to improve its performance. Because of their exceptional performance at both high and low temperatures and their resilience to fatigue, styrene butadiene styrene (SBS)-modified asphalt binders are employed in a significant portion of asphalt pavement construction projects across the world [3].

Although research on the current asphalt thermal aging mechanism is fairly advanced [4], because road workers do not pay attention to asphalt ultraviolet aging, research progress is slow. This is because the thermal aging of asphalt and ultraviolet aging incentives are entirely different. Because the UV aging study of asphalt plays an important guiding role in building roads in western China, several road researchers have recently begun to pay attention to the effects of UV aging on asphalt mixes and asphalt [5]. Several researchers have found that many factors affect the properties of asphalt during aging [6,7,8,9]. Yuanyuan Li et al. pointed out that UV aging has a significant effect on the service life of asphalt pavements due to the fact that the increased penetration of UV light during UV aging increases the UV aging depth [10]. Zhang Hailin found that when studying the changes in elemental composition and rheological properties of asphalt before and after thermal and UV aging using elemental analysis and DSR tests, UV aging resulted in greater changes in indices and more aging [11]. Lixing Ma et al. found that the water stability performance of asphalt mixes decayed as the UV aging time increased [12]. Mona Nobakht et al. demonstrated that intense UV exposure reduces the shear strength of asphalt, and that the low-temperature cracking and fatigue properties of asphalt decrease with aging time [13]. Several people have begun to work on the mechanism of UV aging of asphalt and improvement methods, since the risks of UV aging have been made clear through the tests of some road researchers. Zheng et al. used permeability, viscosity, and ductility to develop a nonlinear equation for the decay of asphalt properties after UV aging [14]. Min. Xiao simulated the dynamic behavior of asphalt microstructure during UV aging using Materials Studio software, a process different from the internal thermal aging mechanism of asphalt [15]. Feng Zhengang et al. found that the addition of UV absorber UV-531 can improve the high and low temperature performance of asphalt in addition to effectively improving the anti-UV aging ability of asphalt [16]. Lingling Hong found that adding UV-531 to SBS-modified asphalt with block ratios of 40/60 and 30/70 could effectively improve its high- and low-temperature properties [17]. UV-531 provides superior performance, is the most widely used ultraviolet absorber, can powerfully absorb the wavelength of 300–345 nm ultraviolet light, and is one of the most used UV absorbers by road researchers to improve the UV aging performance of asphalt materials. Therefore, finding the right dosage is essential to improve the performance and economy of asphalt pavements by improving the UV aging resistance of asphalt.

According to relevant studies, the UV-531 content that can improve the UV aging resistance of asphalt is in the range of 0.2% to 1.2% [15]. Hence, the paper chose three UV-531 doping levels—0.4%, 0.7%, and 1.0%. UV-531 was added to SBS-modified asphalt by melt blending. Needle penetration, softening point, and 5 °C ductility tests were chosen to assess the effect of UV-531 on the conventional properties of SBS-modified asphalt. UV aging of treated asphalt was performed using DSR and DMA tests to evaluate the effect of UV-531 on the high- and low-temperature performance of SBS-modified asphalt. Finally, the optimum dose was derived after evaluating the UV-531/SBS-modified asphalt for UV aging resistance using rutting factor ratio, viscosity index, and infrared spectroscopy.

## 2. Materials and Methods

### 2.1. Testing Raw Materials

UV absorber: UV-531 (2-Hydroxy-4-n-Octyloxy benzophenone) is a light yellow powder that absorbs ultraviolet rays from sunlight as a light stabilizer. The property indicators are shown in Table 1.

The basic properties of SBS-modified asphalt are shown in Table 2.

### 2.2. UV-531/SBS-Modified Asphalt Preparation

The high-speed shear method was used to prepare UV-531- and SBS-modified asphalts to integrate them fully. The specific preparation procedure is as follows.

(1)Put the SBS-modified asphalt into 160 °C ovens for 30 min and then take it out.(2)0%, 0.4%, 0.7%, and 1.0% of UV-531 (external admixture) were added to SBS-modified asphalt and sheared using a high-speed shear rate of 3000 r/min and a shear temperature controlled at 165 °C to 170 °C.(3)UV-531- and SBS-modified asphalt was mixed and sheared at high speed for 40 min to prepare UV-531/SBS-modified asphalt, blended into the amount of 0% of UV-531 produced that is SBS-modified asphalt (not involving UV absorber).

Figure 1 depicts the UV-531/SBS-modified asphalt preparation procedure. In order to more clearly identify between the four distinct UV-531/SBS-modified asphalt dosages, the produced modified asphalts were given numbers. The numbering outcomes are displayed in Table 3.

### 2.3. Asphalt UV Aging Test Process

The UV aging tests on the prepared modified asphalts were performed in a UV infrared chamber, as shown in Figure 2. According to the existing study [18], asphalt specimens should have a thickness of 1 mm during UV aging; thus, in order to effectively simulate the asphalt film thickness of the in-situ asphalt mixture during the test, the sample container area and the asphalt film thickness (1 mm) were used to calculate the mass of the asphalt UV aging specimens.

The diameter of the glass disc used to hold the asphalt during the test was 14 cm and the thickness of the asphalt film was 1 mm, so 15.4 g of asphalt was weighed into the glass disc. The discs were warmed in the oven at 170 °C until the asphalt was uniformly laid on the surface of the glass discs. Finally, the asphalt aging specimens were placed in the UV aging environmental chamber for 7 d. The UV-aged asphalt specimens are shown in Figure 3, and the process of the asphalt UV aging test is shown in Figure 4.

### 2.4. Tests and Methods

#### 2.4.1. Needle Penetration

The needling values of four SBS-modified asphalts were tested at 25 °C, 100 g, and 5 s using a fully automatic needling instrument. Testing method: dry needle penetration sample dishes are prepared, the asphalt is placed in the vessels cooled to ambient temperature, and then the asphalt vessels are placed in a constant temperature water bath at 25 °C for 1.5 h. Remove the test sample after the water bath, then evaluate the needle penetration of the test sample using the needle penetration tester to determine the needle penetration of the measured data to take the average of three calculations. The resulting value is the needle penetration of modified asphalt.

#### 2.4.2. Softening Point

In this paper, the effect of UV-531 on the high temperature performance of SBS-modified asphalt was evaluated by testing the softening point values of four SBS-modified asphalts using a fully automatic softening point tester. The test procedure is as follows: take a certain amount of modified asphalt and place it in a standard mold, cool the mold together with the internal asphalt to room temperature, and then scrape the test sample using a hot scraper. Place the test sample in a constant temperature bath of 5 ± 0.5 °C for 30 min. Use the softening point tester to test the samples until the test sample wrapped inside the steel ball just falls on the receiving plate when stopped, then record the temperature of the test sample. The temperature data is the softening point of the modified asphalt.

#### 2.4.3. 5 °C Ductility

The authors chose a temperature of 5 °C for the ductility test. The test was performed by slowly pouring the modified asphalt into the mold from one end to the other, allowing the mold and the asphalt inside to cool to room temperature before scraping the test sample with a hot scraper. The test sample was placed in a water bath at a temperature of 5 ± 0.5 °C for 1.5 h. Using a ductility tester, the sample was tested until the asphalt was pulled off and the scale was read, which provided the ductility of the modified asphalt.

#### 2.4.4. High-Temperature Dynamic Shear Rheology Test

Temperature scanning tests were performed on four asphalt specimens using a dynamic shear rheometer (DSR) with a temperature range of 30–100 °C and a frequency of 10 ± 0.1 rad/s. The DSR test method is accomplished by measuring the viscous and elastic properties of a thin asphalt binder specimen sandwiched between an oscillating plate and a fixed plate [19].

The high temperature PG index is determined by the DSR of the unaged asphalt, as reflected in the rutting factor curve, at the temperature corresponding to G*/sinδ ≤ 1.00 kpa [20].

#### 2.4.5. Low-Temperature Glass Transition Temperature Test

The modulus of a viscoelastic material can be tested using the Dynamic Mechanical Analysis (DMA) test as a function of time, temperature, or frequency. Only a tiny sample is needed to ascertain the material’s dynamic mechanical properties over a large range of temperatures or frequencies. Figure 5 and Figure 6 display the clamp and dynamic mechanical analyzer.

The ideal modulus–temperature curve of modified asphalt typically has four regions: the glassy region, the glassy–rubbery transition region, the rubbery plateau region, and the flow region. The service temperature of asphalt mixtures undergoes the entire process from the glassy to the viscous flow state, and the mechanical condition of the asphalt mixture is essentially a macroscopic depiction of molecular mobility. The glass transition temperature can describe the asphalt mixture’s low-temperature performance. When the glass transition temperature is lower than the minimum service temperature and the asphalt can perform as intended, this is the perfect state for the asphalt mixture. This gives the asphalt mixture good deformability throughout the service temperature and can relax the temperature stresses caused by the lowering of the temperature, reducing the generation of low-temperature cracks.

#### 2.4.6. Rutting Factor Ratio

The rutting factor ratio is the ratio of the asphalt rutting factor after aging and before aging. The formula for calculating the rutting factor ratio is shown in Equation (1).
(1)$$\mathrm{TR}=\frac{\left(G_{2}^{*}/\sin\delta_{2}\right)}{\left(G_{1}^{*}/\sin\delta_{1}\right)}$$
where: TR is the rutting factor ratio; *δ*_1_ and *δ*_2_ are the phase angles before and after aging, respectively; $G_{1}^{*}$ is the complex modulus before aging, and $G_{2}^{*}$ is the complex modulus after aging.

#### 2.4.7. Viscosity Aging Index

Asphalt aging, viscosity increases, and the viscosity curve move up a distance, and this distance can be called the viscosity aging index. A lower viscosity aging index indicates superior anti-aging performance, and the viscosity aging index calculation formula is shown in Formula (2).
(2)$$C=\mathrm{lg}\mathrm{lg}(\eta_{a}\times 10^{3})-\mathrm{lg}\mathrm{lg}(\eta_{0}\times 10^{3})$$
where: C is the viscosity aging index, *η*_a_ is the viscosity of the asphalt after aging, and *η*_0_ is the viscosity of the original asphalt.

#### 2.4.8. Infrared Spectroscopy

Infrared spectroscopy is based on the information of atomic vibrations and rotations inside molecules to analyze and determine substance composition and molecular structure [21]. Infrared spectrograms can be applied to the determination of the molecular structure of compounds, identification of unknowns, and analysis of mixture composition [22]. In other words, changes in the compositional makeup of asphalt prior to and subsequent to aging can be studied by infrared spectroscopy.

## 3. Results

### 3.1. Effect of UV-531 on the Conventional Performance of SBS-Modified Asphalt

The effect of UV-531 on the conventional performance of SBS-modified asphalt was investigated by adding UV-531 at 0.4%, 0.7%, and 1.0% to SBS-modified asphalt and testing the needle penetration, softening point, and 5 °C ductility.

#### 3.1.1. Needle Penetration Results

Needle penetration measures asphalt viscosity and is strongly related to asphaltene content; the higher the asphaltene content and the lower the aromatic fraction content, the lower the needle penetration index [23]. Figure 7 depicts the test results.

As can be seen from Figure 7, with the increase of UV-531 admixture, the needle penetration first increased and then decreased, at 0.7% admixture, increased by 24%, overall UV-531/SBS-modified asphalt needle penetration than SBS-modified asphalt slightly increased. Mainly due to the molten state, between the heterogeneous polymer molecules break up and convective agitation, coupled with the strong shear effect of mixing equipment, the formation of a part of the graft or block copolymer, so that the original asphalt intermolecular force is reduced, mobility increased, the needle penetration increased.

#### 3.1.2. Softening Point Results

The softening point of asphalt characterizes its high-temperature performance; the higher the softening point, the better the asphalt’s high-temperature performance. The test results are shown in Figure 8.

Figure 8 indicates that the softening point tended upward when the concentration of UV-531 rose. Although there is a small increase of 4.9% at 1.0% compared to SBS-modified bitumen, this indicates that UV-531 has little effect on the high-temperature properties of SBS-modified bitumen.

#### 3.1.3. 5 °C Ductility Results

Ductility characterizes the low temperature cracking resistance of asphalt. The higher the ductility value, the greater the elastic component and the lower the viscous component, the better the low-temperature crack tolerance of the asphalt. In this study, 5 °C ductility was used to investigate the impact of UV-531 on the low-temperature properties of SBS-modified asphalt, and the test is shown in Figure 9.

Figure 9 illustrates how various UV-531 concentrations significantly impacted the ductility of the SBS-modified asphalt. From 0.4% to 1.0% of UV-531, the ductility of SBS-modified asphalt gradually increased, with an increase of 58.3% at 1.0%, showing that UV-531 improved the ability of SBS-modified asphalt to withstand low-temperature distortion.

### 3.2. Effect of UV-531 on the High and Low-Temperature Performance of SBS-Modified Asphalt before and after UV Aging

#### 3.2.1. High-Temperature Dynamic Shear Rheology Test Results

To evaluate the effect of UV-531 on the high-temperature properties of asphalt, we measured the curves of composite shear modulus and phase angle variation with temperature. The test outcomes are shown in Figure 10, Figure 11, Figure 12 and Figure 13.

Figure 10 shows that the composite modulus curves before UV-aging vary widely until 50 °C, and the composite modulus decreases slightly with increasing UV-531 incorporation and is essentially close after 50 °C. As shown in Figure 11, the composite shear modulus for the four asphalts increased to different degrees after UV aging in comparison with the pre-aging period. After UV aging, pitch molecules absorbed radiation energy, bond energy was broken, free radicals were generated, combined with oxygen, and compound modulus inwardly increased.

The phase angle δ reflects the proportion of the viscous component in the composite modulus [24,25]. A larger δ indicates a higher viscous proportion, and vice versa, a higher elastic proportion. From Figure 12, it can be seen that the addition of UV-531 before aging has a small effect on the stage angle of the low temperature section, and after 50 °C, the stage angle tends to drop with the amount of admixture, indicating that UV-531 can increase elasticity of SBS-modified asphalt. Observing Figure 13, it is evident that the warmer the post-aging temperature is, the lower the phase gradient of UV-531/SBS-modified asphalt is with the rising amount of admixture, compared with the increase of elastic component of SBS-modified asphalt, indicating that the anti-aging properties of SBS-modified asphalt can be enhanced by UV-531. Comparing Figure 12 and Figure 13, it is observed that the overall phase-angle increases upon aging, but the phase-angle change of UV-531/SBS-modified asphalt is smaller than that of SBS-modified asphalt, which means that the viscosity ratio of the four asphalts after aging increases and the incorporation of UV-531 suppresses the increase of the viscous component to some extent.

The results of the high temperature PG test are shown in Figure 14, as can be seen, the PG grade decreases after adding UV-531. The analysis is due to the fact that UV-531 particles added to SBS asphalt are dispersed between asphalt molecules, which hinders the relative movement of asphalt molecular chains and increases the intermolecular reactions. The temperature range of PG ranges between 70–82 °C and the phase angle of the four asphalts is about 60°. The A, B, C three asphalt phase angles are less than the SBS-modified asphalt, the complex modulus is basically the same, and the smaller the phase angle, the smaller the rutting factor. It is consistent with the complex modulus curve and phase angle curve.

#### 3.2.2. Low-Temperature Glass Transition Temperature Test Results

The peak value of the loss modulus determines how hot the asphalt material is at the glass transition temperature of the loss modulus–temperature curve. Mamuye, Y. et al. indicate that the glass transition temperature test method is simple, has a clear physical meaning, correlates well with the measured properties of asphalt mixes, and can be used as an evaluation criterion for the good or bad low-temperature properties of asphalt [26]. DiWang et al. investigated the effects of the glass transition temperature Tg and the use of the modulus shift factor in measuring the rheological properties of an asphalt binder at low temperatures, and the glass transition temperature has been widely used [27]. As a result, we can determine how modified asphalt performs at low-temperatures by comparing the modified asphalt’s glass transition temperature prior to and subsequent to UV aging.

From Figure 15, the glass transition temperature of SBS asphalt was reduced by each admixture before aging, and it was reduced by 54.74% at 1.0% admixture, indicating that the incorporation of UV-531 increased aromatic structure of asphalt, and the aromatic content increased with the admixture, improving the overall low temperature resistance by SBS asphalt. The overall temperature after aging decreases, and reaches its lowest at 0.7%, when the glass transition temperature is −12.29 °C, indicating that the increase of UV-531 doping has resisted the increase of glass transition temperature to a certain extent. At this time, UV-531 is likely to capture the free radicals so that the free radicals cannot combine with oxygen and inhibit the oxidation reaction, thus slowing down the aging. Comparing the glass transition temperature before and after aging, it was discovered that the glass transition temperature of asphalt has increased with aging, indicating that UV-531 and SBS asphalt underwent addition and oxidation reactions during aging, resulting in long-chain compounds and oxides, aromatic and resin turning into asphaltene, and worsening low-temperature performance. After aging at 0.7% doping, the curve’s inflection point demonstrates that a more significant dose is not better. In general, the increase of UV-531 has an inhibitory effect on both the chain breaking and oxidation reactions that occur during asphalt aging.

## 4. Evaluation of UV-531 on the Anti-UV Aging Performance of SBS-Modified Asphalt

### 4.1. Evaluation of the Aging Behavior of Modified Asphalt Based on Rutting Factor Ratio

The size of the rutting factor ratio reflects the extent of asphalt hardening due to aging; the more significant its value, the worse the asphalt’s resistance to short-term aging performance. The paper chose rutting coefficients of 70–88 °C to investigate the aging properties of the four asphalts. Table 4 displays the calculated results. The rutting factor change curve is shown in Figure 16.

From Figure 16, the same trends were observed for all four asphalts, which decreased with increasing temperature, but to different degrees. At the same temperature, the rutting factor ratio decreases with increasing doping, indicating that UV-531 incorporation improves the aging resistance. Among the three curves of A, B, and C, the TR decreases with the increase of doping amount, but after a doping amount of 0.7%, there is almost no change in TR, which indicates that the higher doping amount is not better. Comprehensive analysis shows that the addition of UV-531 can inhibit asphalt aging, and the optimal dose is 0.7%.

### 4.2. Evaluation of Asphalt Aging by Viscosity Aging Index

The aging resistance index of asphalt in this experiment uses the 60 °C viscosity ratio of pre-aging and post-aging asphalt. The aging indices of SBS-modified asphalt and A, B, and C asphalts are listed in Table 5.

From Table 5, there is evidence that prior to aging, the viscosities of A, B, and C are less than those of SBS-modified asphalt, and the reason for this analysis is that UV-531 has little effect upon the branched chain fat composition of road asphalt, but it increases the aromatic composition of asphalt [28], so the viscosity decreases. The overall viscosity of the aged asphalt rises due to the simultaneous oxidative hardening of the matrices and the aging decomposition of the SBS polymer [29], the decrease of the aromatic content, the increase of the carbon-based index, the increase of ashaltene [30], and the increase of molecular volume of asphalt, which causes the asphalt to harden and the visibility to increase. Aging index can be found that UV-531/SBS-modified asphalt is less than SBS-modified asphalt, meaning that UV-531 improves asphalt aging resistance. The aging index at 0.7% admixture is the smallest, indicating that UV-531 admixture has the best content. In general, UV-531 was favorable to improve the aging resistance of asphalt.

### 4.3. Analysis of the Degree of Aging Using Infrared Spectroscopy

Fusong Wang et al. indicated that some double bonds in SBS copolymers degrade during aging and oxidize to oxygenated groups, such as hydroxyl, carbonyl group, and ether bonds [31], carbonyl absorption peak size can reflect the aging degree, the larger the peak, the deeper the aging. This experiment tested SBS-modified asphalt with UV-531 dosed at 1.0% (i.e., C), Figure 17 and Figure 18 show the test results.

As seen in Figure 17 and Figure 18, the characteristic peaks of the infrared spectra changed more obviously in the first region (4000–2800), the second region (2800–1900), and the third region (1900–1300), while the characteristic peaks in the fourth region (1300–500) did not change; only the absorption intensity appeared to be different.

Table 6 shows the wave numbers corresponding to some characteristic peaks at wave numbers 1595.046, 2364.603, and 3705.059 for the -C=O group, -C≡C group, and -OH group, respectively. After UV aging, an apparent -OH (conjoined) absorption peak was observed in the first region near position 3750, indicating the occurrence of chemical reactions that resulted in the production of alcohols, phenols, or organic acids, which should be a component of long-chain compounds in the aging process of chain-breaking reactions, primarily oxidation reactions. At position 2364.603 in the second area, the disappearance of the -C≡C group was discovered, showing that the -C≡C group carbon element was involved in the chemical reaction that produced R’≡R symmetric molecules without an infrared spectral band; the number of carbonyl groups in the aromatic rings of the two asphalts have changed significantly as a result of UV aging, as shown by the third region in 1595.046 position -C=O absorption peak, which becomes more prominent. A larger carbonyl absorption peak denotes more severe UV aging, thus, the SBS-modified asphalt and C asphalt exhibit UV aging ranging from severe to mild.

## 5. Conclusions

In this paper, the needle penetration, softening point, and 5 °C ductility of UV-531/SBS asphalt for the purpose of assessing the effect of UV-531 on the conventional properties of SBS-modified asphalt were measured. DSR and DMA tests were performed to estimate their effects on the high and low temperature properties of SBS-modified bitumen. Three methods were chosen to investigate the aging behavior of modified asphalt UV-531/SBS, such as rutting factor ratio, viscosity aging index, and infrared spectroscopy. In a comprehensive view, the addition of UV-531 can enhance the UV aging resistance of asphalt effectively with an optimal dose of 0.7%. The main conclusions are listed below.

(1)The ductility at 5 °C and needle penetration of SBS-modified asphalt were improved with the increase of UV-531 addition, while the softening point was less affected. This means that the addition of UV-531 improved the low temperature properties and viscosity of SBS-modified asphalt, while the high temperature properties were less affected.(2)The DSR test revealed that UV-531 enhanced the high temperature rheological properties of SBS-modified asphalt to some degree. It was shown in the DMA test that the increase of UV-531 could reduce the glass transition temperature by a maximum of 54.74%, which led to a significant improvement in the low temperature anti-cracking properties of asphalt after ultraviolet light aging.(3)Three evaluation methods were selected to evaluate the aging. In the evaluation method based on the rutting coefficient ratio, the rutting coefficient ratio reduced as the amount of UV-531 doping increased, demonstrating that UV-531 inhibits UV aging. In the evaluation, a method based on viscosity aging index was used. As the viscosity before aging decreased with increasing dosing, the viscosity aging index also decreased, reaching a minimum at a dosing of 0.7%, indicating that UV-531 can increase the aromatic structure of asphalt. According to the analysis of the degree of aging by infrared spectra, the characteristic peaks of infrared spectra vary more obviously in the first, second and third regions, and the analysis results were that the UV aging degree of UV-531/SBS asphalt with 0.7% admixture was significantly lower than that of SBS-modified asphalt.

## Figures and Tables

**Figure 1 materials-15-08110-f001:**
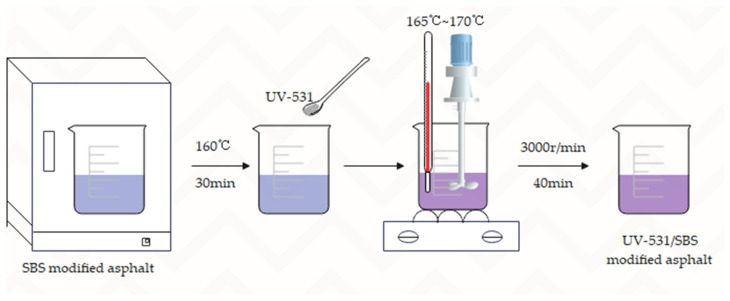
Flow chart of UV-531/SBS asphalt preparation.

**Figure 2 materials-15-08110-f002:**
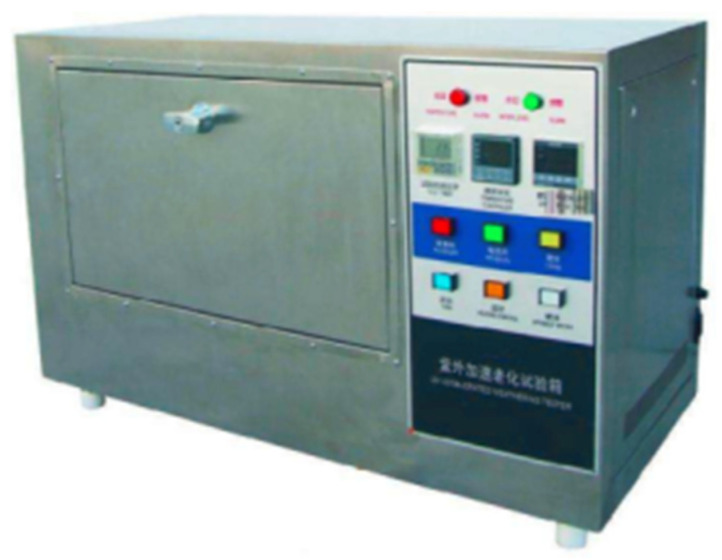
UV aging environment box for asphalt.

**Figure 3 materials-15-08110-f003:**
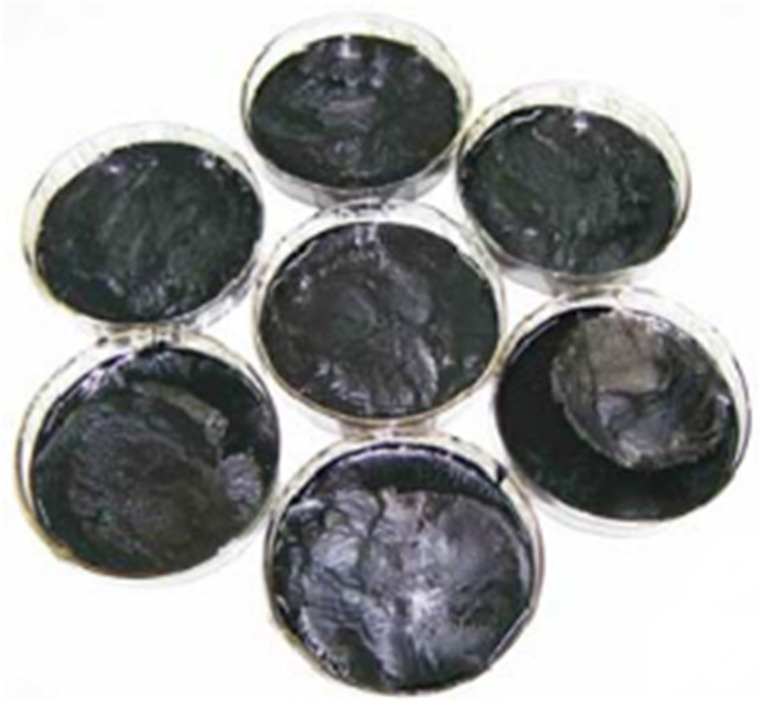
Aging asphalt sample.

**Figure 4 materials-15-08110-f004:**
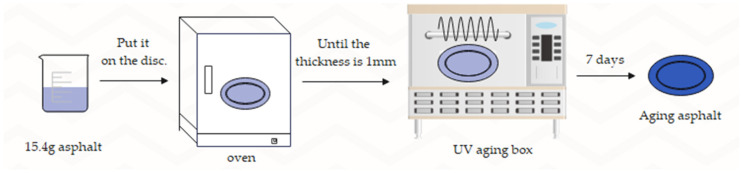
Ultraviolet aging experiment diagram.

**Figure 5 materials-15-08110-f005:**
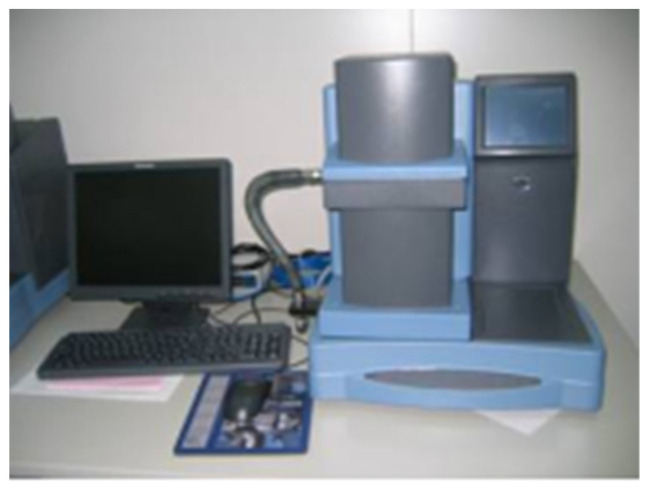
Dynamic Mechanical Analysis.

**Figure 6 materials-15-08110-f006:**
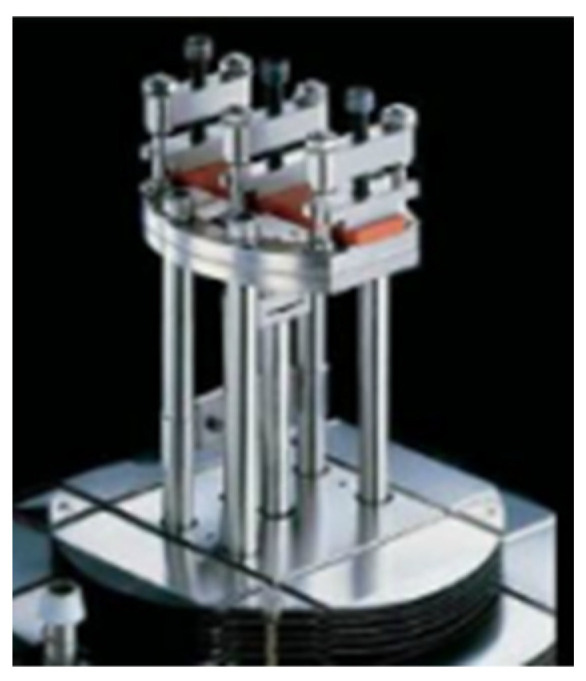
The clamp of Dynamic Mechanical Analysis.

**Figure 7 materials-15-08110-f007:**
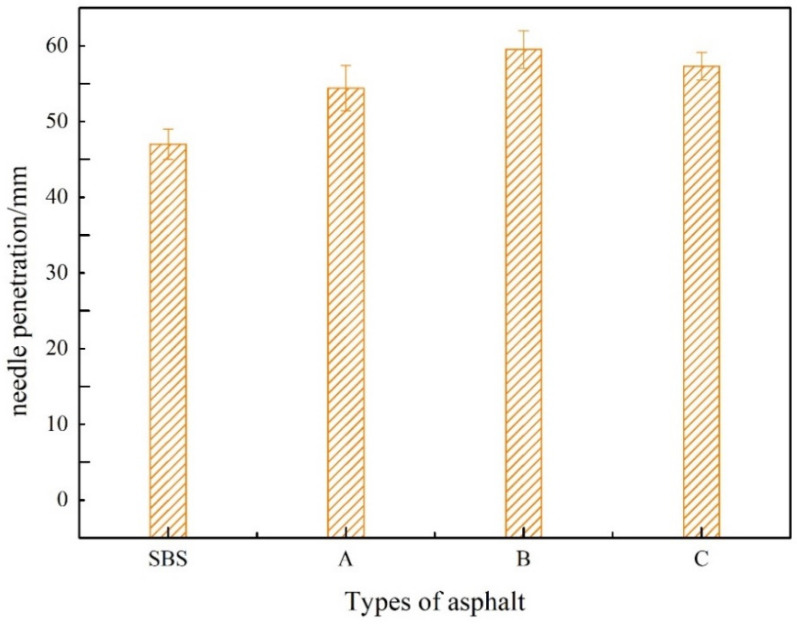
Needle Penetration of Modified Asphalt at 25 °C.

**Figure 8 materials-15-08110-f008:**
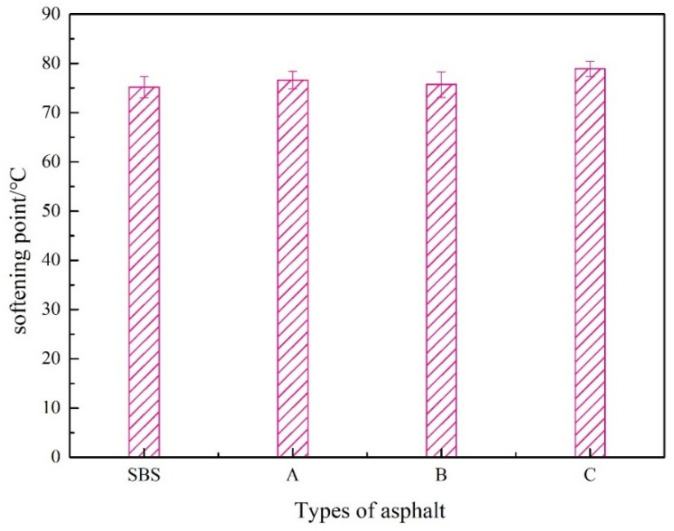
Softening Point of Modified Asphalt.

**Figure 9 materials-15-08110-f009:**
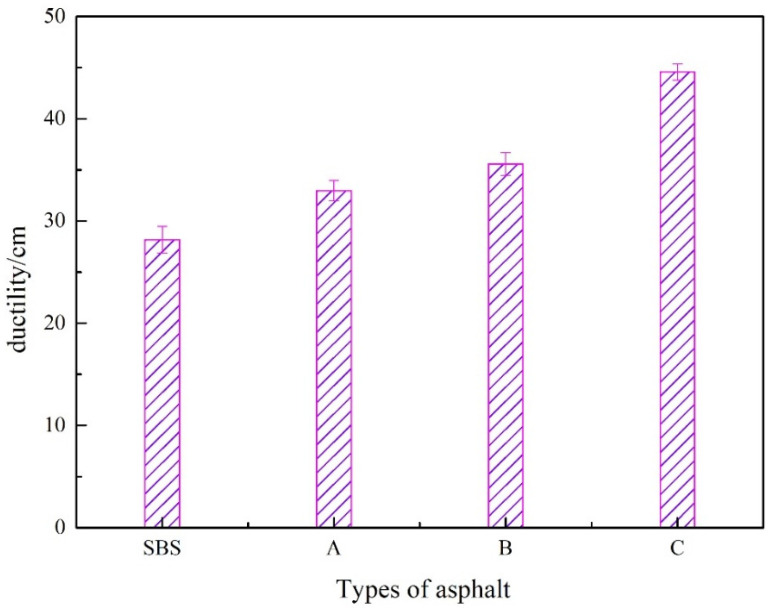
Ductility of Modified Asphalt at 5 °C.

**Figure 10 materials-15-08110-f010:**
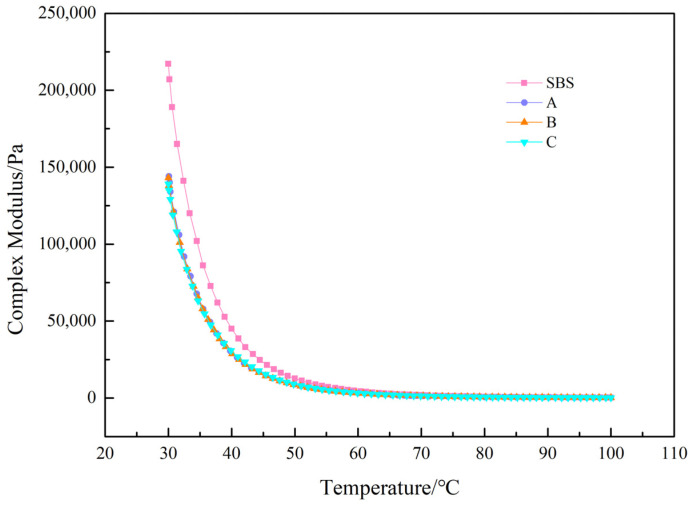
Modified asphalt complex modulus before UV aging.

**Figure 11 materials-15-08110-f011:**
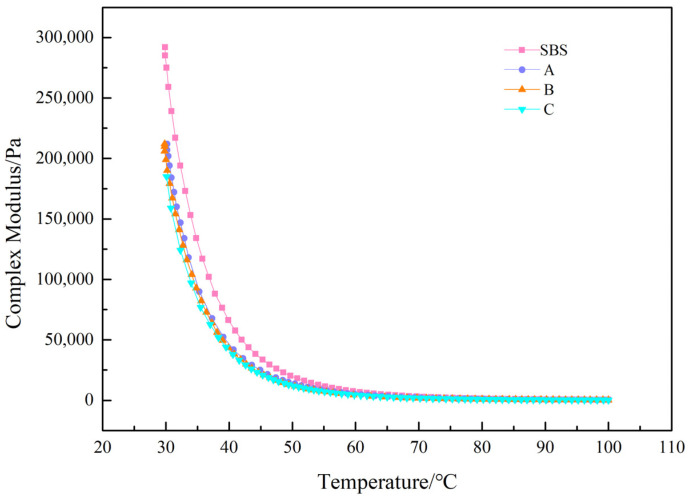
Modified asphalt complex modulus after UV aging.

**Figure 12 materials-15-08110-f012:**
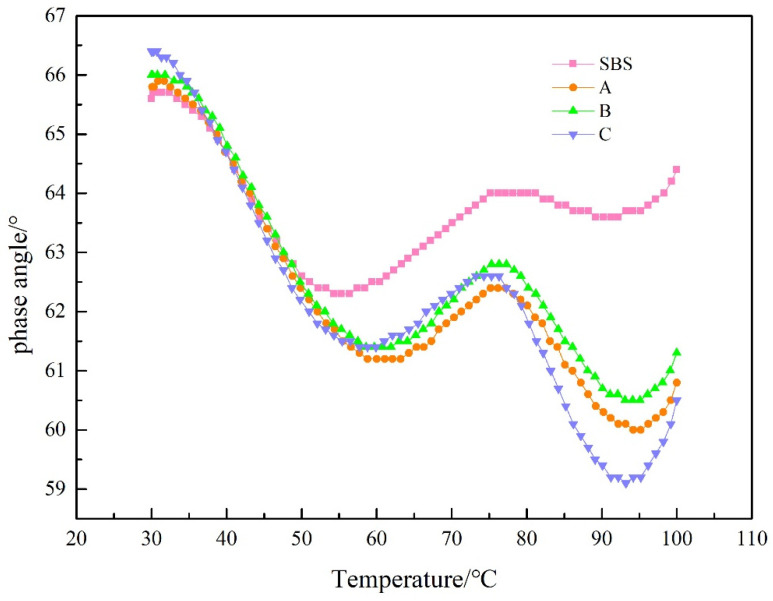
The phase angle of modified asphalt before UV aging.

**Figure 13 materials-15-08110-f013:**
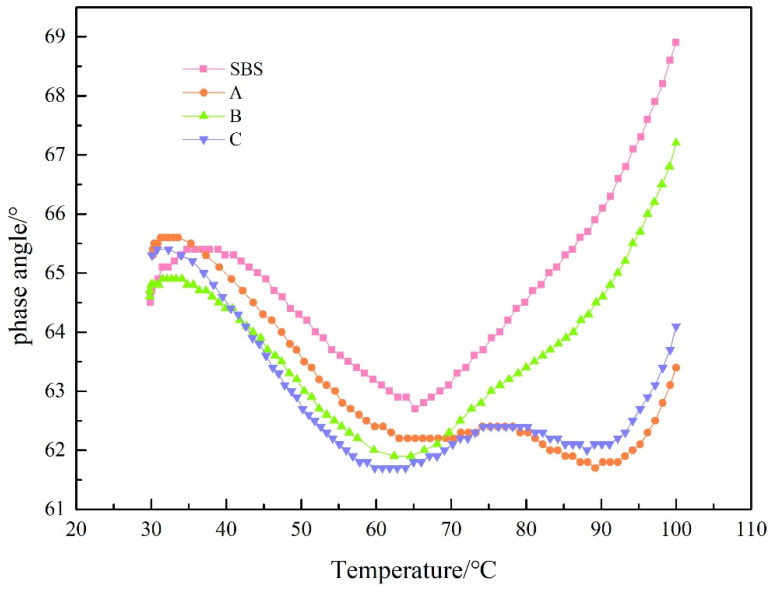
The phase angle of modified asphalt after UV aging.

**Figure 14 materials-15-08110-f014:**
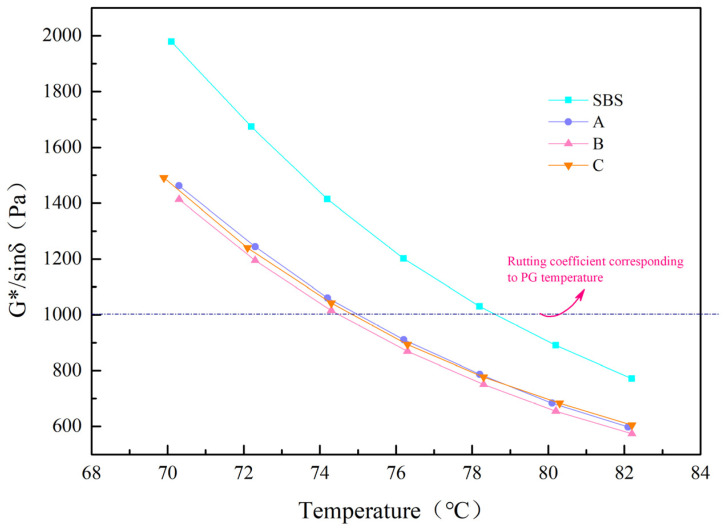
Rutting coefficient of each sample at different temperatures.

**Figure 15 materials-15-08110-f015:**
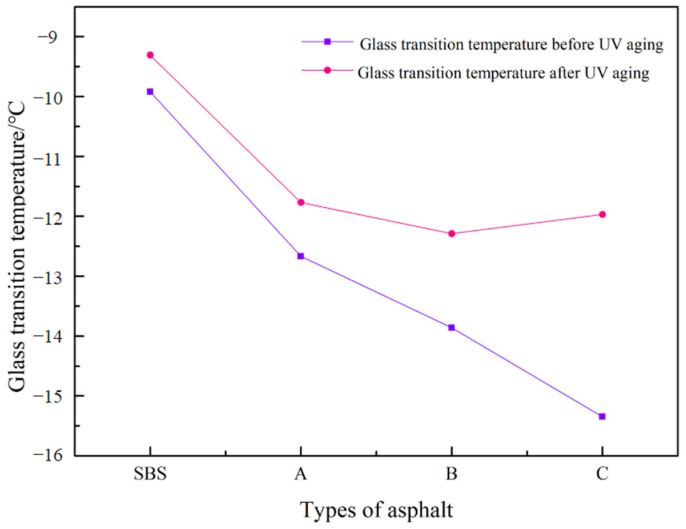
Glass Transition Temperature of Modified Asphalt.

**Figure 16 materials-15-08110-f016:**
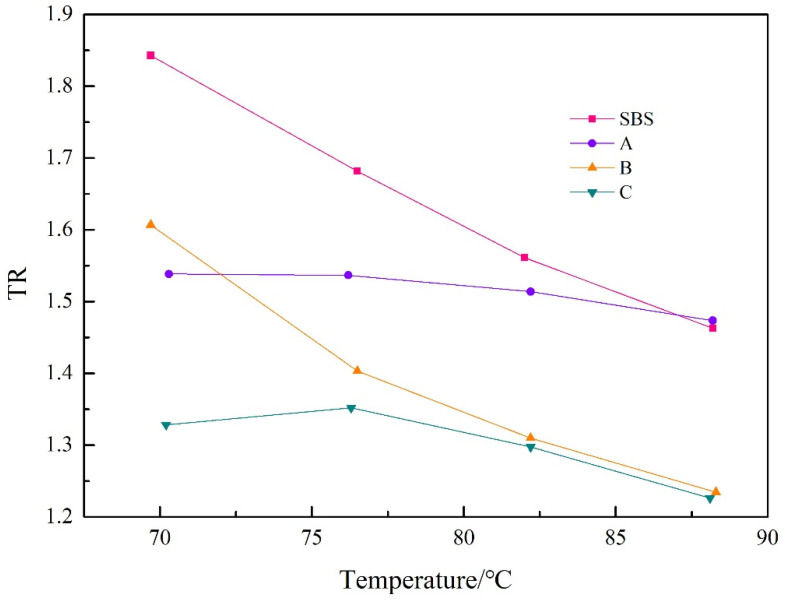
The G*/sinδ ratio before and after aging of SBS/A/B and C.

**Figure 17 materials-15-08110-f017:**
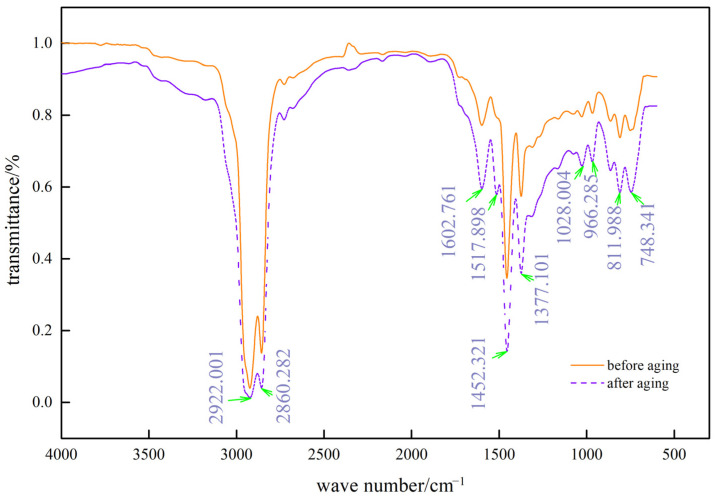
Infrared spectra of SBS-modified asphalt before and after UV aging.

**Figure 18 materials-15-08110-f018:**
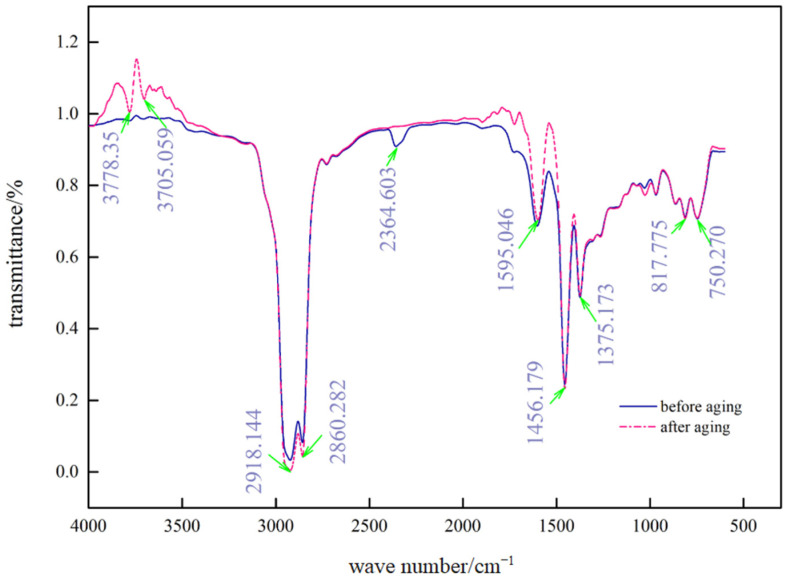
Infrared spectra of C asphalt before and after UV aging.

**Table 1 materials-15-08110-t001:** Performance index of UV-531.

Indicators	Technical Requirements	Indicators	Technical Requirements
Density/g/cm^3^	1.068	Boiling/°C	457.9
Melting point/°C	47~49	Flash point/°C	155.1

**Table 2 materials-15-08110-t002:** Fundamental characteristics of SBS-modified asphalt.

Test Items		SBS-Modified Asphalt	
		Real Test	Technical Requirements
Ductility (5 °C, 5 cm/min)/cm Not less than		28.0	20
Softening point (Global Law)/°C Not less than		74.2	60
Needle penetration (25 °C, 100 g, 5 s)/0.1 mm		53.0	40~60
Resilient Recovery (25 °C)/% Not less than		94.0	75
Flash Point/°C Not less than		300	230
Needle penetration index PI Not less than		0.26	0
Kinematic viscosity (135 °C)/Pa·s No greater than		2.55	3.0
Solubility (trichloroethylene)/% Not less than		99.7	99
After RTFOT test	Quality change/% No greater than	−0.008	±1.0
	Ductility (5 °C)/cm Not less than	19.7	15
	Needle penetration ratio (25 °C)/% Not less than	83.2	65

**Table 3 materials-15-08110-t003:** Modified Asphalt Number.

Modified Asphalt	Number
SBS-Modified Asphalt	SBS
0.4%UV-531 + SBS-Modified Asphalt	A
0.7%UV-531 + SBS-Modified Asphalt	B
1.0%UV-531 + SBS-Modified Asphalt	C

**Table 4 materials-15-08110-t004:** The G*/sinδ ratio before and after aging of SBS/A/B and C.

Temperature (°C)	SBS			A			B			C		
	$G_{1}^{*}/\sin\delta_{1}$	$G_{2}^{*}/\sin\delta_{1}$	TR	$G_{1}^{*}/\sin\delta_{1}$	$G_{2}^{*}/\sin\delta_{1}$	TR	$G_{1}^{*}/\sin\delta_{1}$	$G_{2}^{*}/\sin\delta_{1}$	TR	$G_{1}^{*}/\sin\delta_{1}$	$G_{2}^{*}/\sin\delta_{1}$	TR
70	1977	3714	1.87	1462	2249	1.53	1413	2536	1.79	1490	1980	1.32
76	1201	2048	1.71	910	1399	1.54	869	1245	1.43	893	1207	1.35
82	771	1204	1.56	598	905	1.51	573	751	1.31	604	783	1.29
88	515	753	1.46	412	607	1.47	395	488	1.23	428	525	1.22

**Table 5 materials-15-08110-t005:** The aging index of SBS, A, B and C modified asphalt at 60 °C.

Asphalt Type	Original *η_a_*	Aging *η_a_*	C
SBS-modified asphalt	773.918	1275.485	0.01571
A	608.138	910.510	0.01297
B	596.864	767.58	0.00814
C	610.044	822.675	0.00964

**Table 6 materials-15-08110-t006:** Characteristic peak distribution table.

**Wave Number/cm^−1^**	1595.046	2364.603	3705.059
**Feature Peak**	-C=O	-C≡C	-OH group of alcohols and phenols

## Data Availability

Not applicable.

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
