# Peer review of "Study on the Effect of Ultraviolet Absorber UV-531 on the Performance of SBS-Modified Asphalt"

_materials, 2022, doi:10.3390/ma15228110_

Round 1

Reviewer 1 Report

The paper is well structured and adequately divided into sections and sub-sections and the manuscript reports a substantial body of results and seems a well-done job but I have some recommendations before accepting this manuscript. 

- The bibliography should be actualized. Most of the cited papers are from last year and only one is dated 2022.

- In section 2 Materials and Methods please describe all the tests and methods utilized for material characterization. 

please check formula number 2. Do you have 2 lg there? 

In my opinion, it is essential that the authors consult the bibliography from the last year in order to prove the introduction of the article. Either in this manuscript, the information is at the level of 2021 or 2020. Also, the authors must describe the methods used to obtain the results. Otherwise, I cannot verify their correlation. I requested this clearly from the authors. Also, the formula I refer to contains the logarithm of the logarithm. How do I know if it's a typing error or if it was actually mathematically applied like that? In this case, the calculations may be distorted.

Reviewer 2 Report

The reviewer appreciates the effot given in the manuscript. However, there are some improvements and revisions which have to be handled by authors:

- The manuscript is in need of proofreading. Please check and revise the manuscript by considering the typo and grammer mistakes. 

-It is suggested that SBS modified samples not involving UV absorber should be investigated and the performance result of the abovementioned sample should be compared with the ones involving different amount of UV absorber.

-Did you consider the effects of mixing conditions on the SBS modified sample involving UV absorber? Mixing for 40 mins by 3000 r/min shearing rates will cause some more aging on the SBS modified samples. For this reason, it is required that the SBS modified samples (not involving UV absorber) should be subjected to the same mixing conditions. 

- Please revise the subtitle (line 141)

-Line 143: please mention another term instead of "larger ductility".

-Please indicate the frequency used for the DSR testing.

- Regarding to DSR testing, it is recommended to investigate more rheological characteristics as PG upper limit etc. According to the frequency used for the testing, it is simple to find the PG upper temperature of the sample. Please see the paper below: 

"Özdemir, D. K. (2021). High and low temperature rheological characteristics of linear alkyl benzene sulfonic acid modified bitumen. Construction and Building Materials, 301, 124041."

-How did you decide on the TR temperature?

-Why did you only investigate the sample C in FTIR analysis? Additionally, it is recommended to present the FTIR results in a table, which would be more clear and easy to follow. 

Round 2

Reviewer 1 Report

The authors have addressed all the queries. The article may be accepted in the present form. 

Reviewer 2 Report

Thank you for your revisions/answers.